# DEFT: Dexterous Fine-Tuning for Hand Policies

**Aditya Kannan**[*]    **Kenneth Shaw**[*]    **Shikhar Bahl**
**Pragna Mannam**    **Deepak Pathak**

Carnegie Mellon University

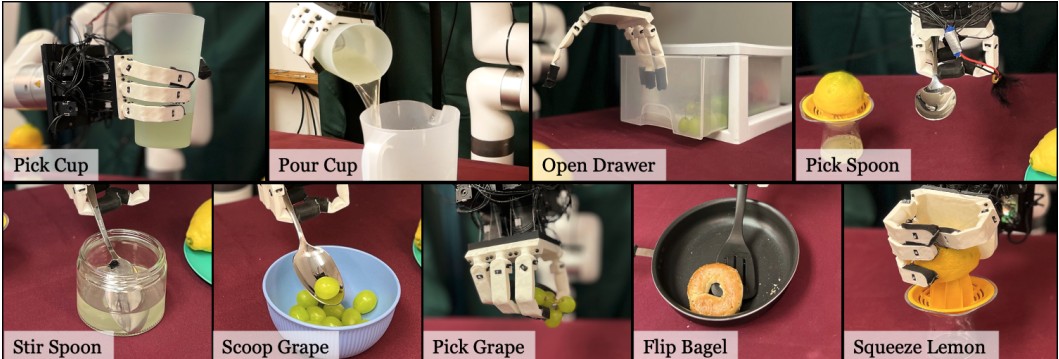

Figure 1: We present DEFT, a novel approach that can learn complex, dexterous tasks in the real world in an efficient manner. DEFT manipulates tools and soft objects without any robot demonstrations.

**Abstract:** Dexterity is often seen as a cornerstone of complex manipulation. Humans are able to perform a host of skills with their hands, from making food to operating tools. In this paper, we investigate these challenges, especially in the case of soft, deformable objects as well as complex, relatively long-horizon tasks. However, learning such behaviors from scratch can be data inefficient. To circumvent this, we propose a novel approach, DEFT (**DE**xterous **F**ine-**T**uning for Hand Policies), that leverages human-driven priors, which are executed directly in the real world. In order to *improve* upon these priors, DEFT involves an efficient online optimization procedure. With the integration of human-based learning and online fine-tuning, coupled with a soft robotic hand, DEFT demonstrates success across various tasks, establishing a robust, data-efficient pathway toward general dexterous manipulation. Please see our website at `https://dexterous-finetuning.github.io` for video results.

**Keywords:** Dexterous Manipulation, Reinforcement Learning, Learning from Videos

## 1    Fine-Tuning Affordance for Dexterity

The goal of DEFT is to learn useful, dexterous manipulation in the real world that can generalize to many objects and scenarios. DEFT learns in the real-world and fine-tunes robot hand-to-object interaction in the real world using only a few samples. However, without any priors on useful behavior, the robot will explore inefficiently. Especially with a high-dimensional robotic hand, we need a strong prior to effectively explore the real world. We thus train an affordance model on human videos that leverages human behavior to learn what are reasonable behaviors the robot should perform.

### 1.1    Learning grasping affordances

To learn from dexterous interaction in a sample efficient way, we use human hand motion as a prior for robot hand motion. We aim to answer the following: (1) What useful, actionable information can we extract from the human videos? (2) How can human motion be translated to the robot embodiment to guide the robot? In internet videos, humans frequently interact with a wide variety of objects. This

7th Conference on Robot Learning (CoRL 2023), Atlanta, USA.

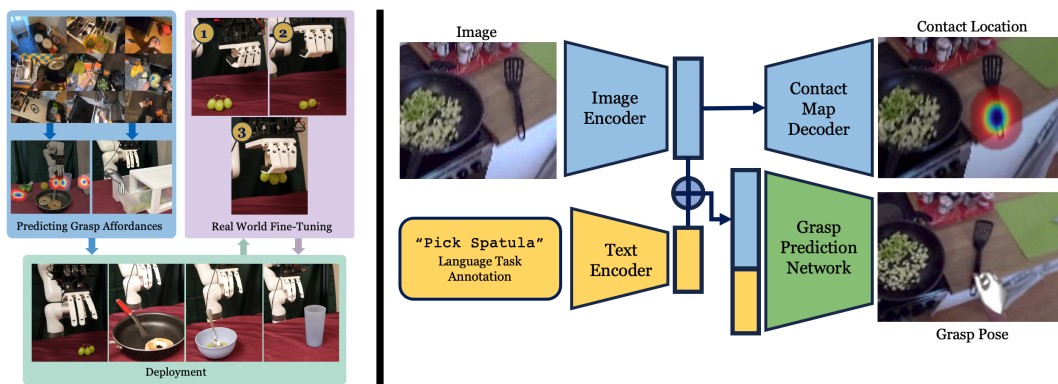

Figure 2: **Left:** DEFT consists of two phases: an affordance model that predicts grasp parameters followed by online fine-tuning with CEM. **Right:** Our affordance prediction setup predicts grasp location and pose.

data is especially useful in learning object affordances. Furthermore, one of the major obstacles in manipulating objects with few samples is accurately grasping the object. A model that can perform a strong grasp must learn *where* and *how* to grasp. Additionally, the task objective is important in determining object affordances–humans often grasp objects in different ways depending on their goal. Therefore, we extract three items from human videos: the grasp location, human grasp pose, and task.

Given a video clip $V = \{v_1, v_2, \ldots, v_T\}$, the first frame $v_t$ where the hand touches the object is found using a pre-trained, off-the-shelf hand-object detection model [1]. Similar to previous approaches [2, 3, 4, 5], a set of contact points are extracted to fit a Gaussian Mixture Model (GMM) with centers $\mu = \{\mu_1, \mu_2, \ldots, \mu_k\}$. Detic [6] is used to obtain a cropped image $v_1'$ containing just the object in the initial frame $v_1$ to condition the model. We use Frankmocap [7] to extract the hand grasp pose $P$ in the contact frame $v_t$ as MANO parameters. We also obtain the wrist orientation $\theta_{\text{wrist}}$ in the camera frame. This guides our prior to output wrist rotations and hand joint angles that produce a stable grasp. Finally, we acquire a text description $T$ describing the action occurring in $V$.

We extract affordances from three large-scale, egocentric datasets: Ego4D [8] for its large scale and the variety of different scenarios depicted, HOI4D [9] for high-quality human-object interactions, and EPIC Kitchens [10] for its focus on kitchen tasks similar to our robot's. We learn a task-conditioned affordance model $f$ that produces $(\hat{\mu}, \hat{\theta}_{\text{wrist}}, \hat{P}) = f(v_1', T)$. We predict $\hat{\mu}$ in similar fashion to [2]. First, we use a pre-trained visual model [11] to encode $v_1'$ into a latent vector $z_v$. Then we pass $z_v$ through a set of deconvolutional layers to get a heatmap and use a spatial softmax to estimate $\hat{\mu}$.

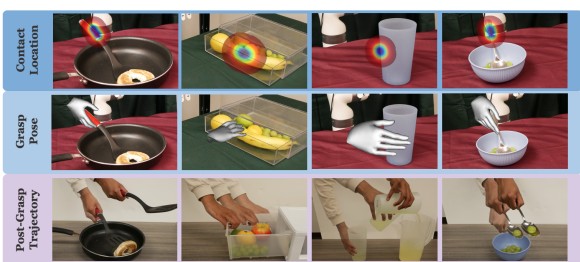

Figure 3: We produce three priors from human videos: the contact location (**top row**) and grasp pose (**middle row**) from the affordance prior; the post-grasp trajectory (**bottom row**) from a human demonstration of the task.

To determine $\hat{\theta}_{\text{wrist}}$ and $\hat{P}$, we use $z_v$ and an embedding of the text description $z_T = g(T)$, where $g$ is the CLIP text encoder [12]. Because transformers have seen success in encoding various multiple modes of input, we use a transformer encoder $\mathcal{T}$ to predict $\hat{\theta}_{\text{wrist}}, \hat{P} = \mathcal{T}(z_v, z_T)$.

At test time, we generate a crop of the object using Segment-Anything [13] and give our model a task description. The model generates contact points on the object, and we take the average as our contact point. Using a depth camera, we can determine the 3D contact point to navigate to. While the model outputs MANO parameters [14] that are designed to describe human hand joints, we retarget these values to produce similar grasping poses on our robot hand in a similar manner to previous approaches [15, 16]. For more details, we refer readers to the appendix.

## 1.2 Fine-tuning via Interaction

**Algorithm 1** Fine-Tuning Procedure for DEFT

---

**Require:** Task-conditioned affordance model $f$, task description $T$, post-grasp trajectory $\tau$, parameter distribution $\mathcal{D}$, residual cVAE policy $\pi$. $E$ number of elites, $M$ number of warm-up episodes, $N$ total iterations.

$\mathcal{D} \leftarrow \mathcal{N}(\mathbf{0}, \sigma^2)$
**for** $k = 1 \ldots N$ **do**
   $I_{k,0} \leftarrow$ initial image
   $\xi_k \leftarrow f(I_{k,0}, T)$
   Sample $\epsilon_k \sim D$
   Execute grasp from $\xi_k + \epsilon_k$, then trajectory $\tau$
   Collect reward $R_k$; reset environment
   **if** $k > M$ **then**
      Order traj indices $i_1, i_2, \ldots, i_k$ based on rewards
      $\Omega \leftarrow \{\epsilon_{i_1}, \epsilon_{i_2}, \ldots, \epsilon_{i_E}\}$
      Fit $\mathcal{D}$ to distribution of residuals in $\Omega$
   **end if**
**end for**
Fit $\pi(.)$ as a VAE to $\Omega$

---

Residual policies have been used previously to efficiently explore in the real world [17, 18]. They use the prior as a starting point and explore nearby. Let the grasp location, wrist rotation and grasp pose, as well as the trajectory from our affordance prior be $\xi$. During training we sample noise $\epsilon \sim \mathcal{D}$ where $\mathcal{D}$ is initialized to $\mathcal{N}(0, \sigma^2)$ (for a small $\sigma$). We rollout a trajectory parameterized by $\xi + \epsilon$. We collect $R_i$, the reward for each $\xi_i = f(v_i) + \epsilon_i$ where $v_i$ is the image. First, we execute an initial number of $M$ warmup episodes with actions sampled from $\mathcal{D}$, recording a reward $R_i$ based on how well the trajectory completes the task. For each episode afterward, we rank the prior episodes based on the reward $R_i$ and extract the sampled noise from the episodes with the highest reward (the 'elites' $\Omega$). We fit $\mathcal{D}$ to the elite episodes to improve the sampled noise. Then we sample actions from $\mathcal{D}$, execute the episode, and record the reward. By repeating this process we can gradually narrow the distribution around the desired values. In practice, we use $M = 10$ warmup episodes and a total of $N = 30$ episodes total for each task. This procedure is shown in Algorithm 1.

At test time, we could take the mean values of the top $N$ trajectories for the rollout policy. However, this does not account for the appearance of different objects, previously unseen object configurations, or other properties in the environment. To generalize to different initializations, we train a VAE [19, 20, 21, 22] to output residuals $\delta_j$ conditioned on an encoding of the initial image $\phi(I_{j,0})$ and affordance model outputs $\xi_j$ from the top ten trajectories. We train an encoder $q(z|\delta_j, c_j)$ where $c_j = (\phi(I_{j,0}), \xi_j)$, as well as a decoder $p(\delta_j|z, c_j)$ that learns to reconstruct residuals $\delta_j$. At test time, our residual policy $\pi(I_0, \xi)$ samples the latent $z \sim \mathcal{N}(\mathbf{0}, \mathbf{I})$ and predicts $\hat{\delta} = p(z, (I_0, \xi))$. Then we rollout the trajectory determined by the parameters $\xi + \hat{\delta}$. Because the VAE is conditioned on the initial image, we generalize to different locations and configurations of the object.

## 2 Experiment Setup

We perform a variety of experiments to answer the following: 1) How well can DEFT learn and improve in the real world? 2) How good is our affordance model? 3) How can the experience collected by DEFT be distilled into a policy? 4) How can DEFT be used for complex, soft object manipulation? Please see our website at `http://dexterous-finetuning.github.io` for videos.

| Method | Pick cup | | Pour cup | | Open drawer | | Pick spoon | | Scoop Grape | | Stir Spoon | |
|---|---|---|---|---|---|---|---|---|---|---|---|---|
| | train | test | train | test | train | test | train | test | train | test | train | test |
| Real-World Only | 0.0 | 0.1 | 0.2 | 0.1 | 0.1 | 0.0 | 0.7 | 0.3 | 0.0 | 0.0 | 0.3 | 0.0 |
| Affordance Model Only | 0.1 | | 0.4 | | **0.5** | | 0.5 | | 0.0 | | 0.3 | |
| DEFT | **0.8** | **0.8** | **0.8** | **0.9** | **0.5** | **0.4** | **0.8** | **0.6** | **0.7** | **0.3** | **0.8** | **0.5** |

Table 1: We present the results of our method as well as compare them to other baselines: Real-world learning without internet priors used as guidance and the affordance model outputs without real-world learning. We evaluate the success of the methods on the tasks over 10 trials.

| Method | Pour Cup | | Open Drawer | | Pick Spoon | |
|---|---|---|---|---|---|---|
| | train | test | train | test | train | test |
| *Reward Function:* | | | | | | |
| R3M Reward | 0.0 | 0.0 | 0.4 | **0.5** | 0.5 | 0.4 |
| Resnet18 Imagenet Reward | 0.1 | 0.2 | 0.3 | 0.1 | 0.4 | 0.2 |
| *Policy Ablation:* | | | | | | |
| DEFT w/ MLP | 0.0 | 0.0 | 0.5 | 0.0 | 0.6 | 0.5 |
| DEFT w/ Transformer | 0.4 | 0.5 | **0.6** | 0.1 | 0.4 | 0.5 |
| DEFT w/ Direct Parameter est. | 0.1 | 0.1 | 0.1 | 0.0 | 0.3 | 0.0 |
| DEFT | **0.8** | **0.9** | 0.5 | 0.4 | **0.8** | **0.6** |

Table 2: Ablations for (1) reward function type, (2) model architecture, and (3) parameter estimation.

## 3   Results

**Effect of affordance model**   We investigate the role of the affordance model and real-world fine-tuning (Table 1 as well as Figure 4). In the real-world only model, we manually provide a few heuristics in place of the affordance prior. We detect the object in the scene using a popular object detection model [13] and let the contact location prior be the center of the bounding box and randomly sample the rotation angle, and a half-closed hand as the grasp pose prior. With these manually provided priors, the robot has difficulty finding stable grasps. Additionally, the main challenge was finding the correct rotation angle for the hand. Hand rotation is very important for many tool manipulation tasks because it requires not only picking the tool but also stable grasping.

**Zero-shot model execution**   We explore the zero-shot performance of our prior. Without applying any online fine-tuning to our affordance model, we rollout the trajectory parameterized by the prior. While our model is decent on simpler tasks, the model struggles on tasks like stir and scoop that require strong, power grasps (shown in Table 1). In these tasks, the spoon collides with other objects, so fine-tuning the prior to hold the back of the spoon is important in maintaining a reliable grip throughout the post-grasp motion. Because DEFT incorporates real-world experience with the prior, it is able to sample contact locations and grasp rotations that can better execute the task.

**Human and automated rewards**   We ablate the reward function used to evaluate episodes. Our method queries the operator during the task reset process to assign a continuous score from 0 to 1 for the grasp. Because the reset process requires a human-in-the-loop regardless, this adds little marginal cost for the operator. But what if we would like these rewards to be calculated autonomously? We use the final image collected in the single post-grasp human demonstration from Section 1 as the goal image. We define the reward to be the negative embedding distance between the final image of the episode and the goal image with either an R3M [11] or a ResNet [23] encoder. The model learned from ranking trajectories with R3M reward is competitive with DEFT in all but one task, indicating that using a visual reward model can provide reasonable results compared to human rewards.

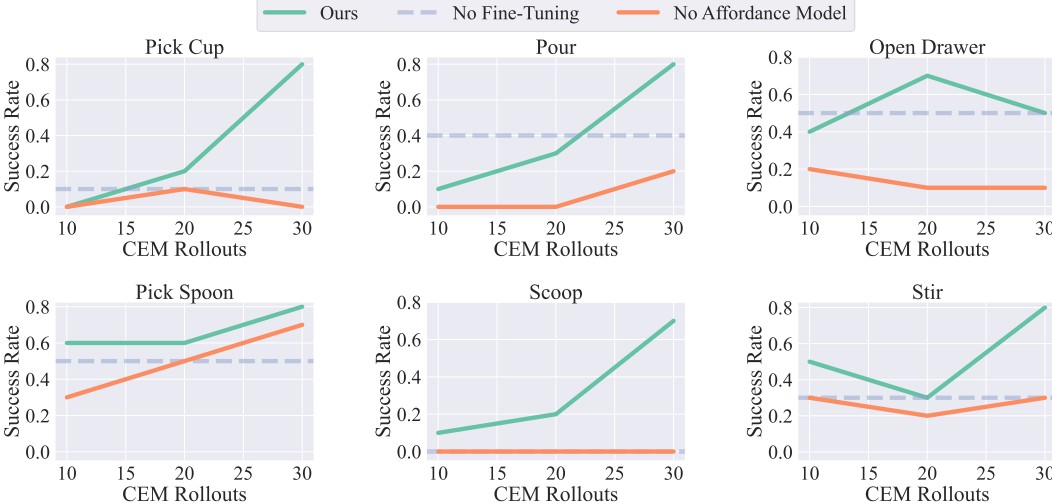

Figure 4: Improvement results for 6 tasks: pick cup, pour, open drawer, pick spoon, scoop, and stir. We see a steady improvement in our method as more CEM episodes are collected.

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
