# OpenReview forum: "DEFT: Dexterous Fine-Tuning for Hand Policies"
_robot-learning.org/CoRL/2023/Workshop/OOD — OOD Workshop @ CoRL 2023_

### Official Review · Reviewer_uu2R · 2023-10-13
**Important work but needs clarifications**

**Rating:** 7
**Confidence:** 5

**Review:**

The paper proposes learning human prior for dexterous hand policies from large-scale human video dataset, and then fine-tuning the policies with CEM on the real setup. The policy transfers well to six manipulation policies considered. A human operator is required to label the task success while real data is collected for fine-tuning.

The work is very relevant to generalizing to novel environments, with fine-tuning on a relatively small number of rollouts (10-30 in reality). The individual components are well known techniques in the literature, but the authors nicely combine them for the challenging tasks.

To be honest, I am a bit surprised that the CEM works so well with a relatively small number of rollouts. One important design choice, the variance of the noise added to exploration, $\sigma$, is not discussed sufficiently in the text. Is the noise being added to each action along the whole trajectory? I understand that the grasp pose is easily perturbed, but seems challenging for the trajectory.

---

### Official Review · Reviewer_ewiW · 2023-10-16
**Review of DEFT**

**Rating:** 7
**Confidence:** 4

**Review:**

**Summary:** This paper proposes a method for efficient learning of dexterous manipulation policies that can generalize to various real-world grasping and manipulation tasks. Specifically, the method proposes a two stage training pipeline, whereby an affordance model prior is trained from large-scale offline human video datasets, and residual policies are learned atop the affordance priors with thirty robot trials in the real world.

**Relevance:** Generalization of dexterous manipulation to new real-world scenarios is challenging and of interest to the community. In the context of this workshop, the paper most strongly relates to 4) “How can we efficiently collect data during deployment and develop learning algorithms that maximally improve system robustness and performance?”

**Novelty:** The method proposes some novel ideas, particularly in the fine-tuning stage, where a residual policy in the form of a VAE is trained from only a handful of real-world episodes.

**Significance:** The results provide insight on comparative tradeoffs of offline training and online fine-tuning. The results demonstrate superior performance compared to directly using manipulation priors learned from human videos or training from scratch in the real-world. The reviewer appreciates the ablations conducted, which compare the performance across policy architectures and automated reward functions.

**Technical strengths and weaknesses:** The approach is sound in the sense that zero-shot generalization from human videos may be wishful thinking, yet improving priors with efficient and generalizable fine-tuning in the real-world offers several advantages. A potential weakness of the method is that its performance appears to rely quite heavily on hand-crafted reward functions for the real-world fine-tuning phase. Crafting these reward functions may become increasingly difficult for complex tasks. Thus, it would be significant if the performance of the method could be improved with the use of automated rewards.

**Quality / clarity:** Overall, the paper is of reasonably high quality and written clearly. There are some sections that could benefit from further details. For example, Section 1.2, which does not appear to indicate that the residual policies are (presumably) fine-tuned for a single dexterous manipulation tasks, and if so, whether the N=30 episodes consist of randomized object positions or if the environment resets deterministically. The images used in Figure 2 are either blurry or too small; this is subject to improvement. There are also some spelling / consistency errors that should be fixed. As examples:
Line 88: “top N trajectories,” should this be top M trajectories?
Lines 93-94: The VAE decoder is mathematically represented differently across these two lines
Line 98: “performs” should be “perform”
Amongst others.

**Recommendation:** Accept.

---

### Decision · Program_Chairs · 2023-10-17

**Decision:**

Accept

**Comment:**

We agree with the reviewers’ assessment that this work is technically sound and will contribute to productive, topical discussions at the 2023 Workshop on OOD Generalization in Robotics. In particular, we appreciate the reviewers' comment that " improving priors with efficient and generalizable fine-tuning in the real-world offers several advantages" to adapt to novel distributions. We recommend the authors incorporate the reviewers’ feedback into their camera-ready submission to further improve their manuscript.